# A recyclable and light-triggered nanofibrous membrane against the emerging fungal pathogen *Candida auris*

Xinyao Liu[1,2,3], Chuan Guo[4], Kaiwen Zhuang[1,2,3], Wei Chen[5], Muqiu Zhang[1,2,3], Yalin Dai[6], Lin Tan [4]*, Yuping Ran [1,2,3]*

**1** Department of Dermatovenereology, West China Hospital, Sichuan University, Chengdu, China,
**2** Laboratory of Dermatology, Clinical Institute of Inflammation and Immunology (CIII), Frontiers Science Center for Disease-related Molecular Network, West China Hospital, Sichuan University, Chengdu, China,
**3** Academician Workstation of Wanqing Liao, West China Hospital, Sichuan University, Chengdu, China,
**4** College of Biomass Science and Engineering, Sichuan University, Chengdu, China, **5** Department of Physics, The University of Texas at Arlington, Arlington, Texas United States of America, **6** Division of Clinical Microbiology, Department of Laboratory Medicine, West China Hospital, Sichuan University, Chengdu, China

* tanlinou@scu.edu.cn (LT); ranyuping@vip.sina.com (YR)

**Data Availability Statement:** All relevant data are within the manuscript and its Supporting Information files.

## Abstract

The emerging "super fungus" *Candida auris* has become an important threat to human health due to its pandrug resistance and high lethality. Therefore, the development of novel antimicrobial strategy is essential. Antimicrobial photodynamic therapy (aPDT) has excellent performance in clinical applications. However, the relevant study on antifungal activity and the mechanism involved against *C. auris* remains scarce. Herein, a recyclable and biodegradable polylactic acid-hypocrellin A (PLA-HA) nanofibrous membrane is newly developed. *In vitro* PLA-HA-aPDT could significantly reduce the survival rate of *C. auris* plankton and its biofilms, and the fungicidal effect of the membrane is still significant after four repeated uses. Simultaneously, PLA-HA exhibits good biocompatibility and low hemolysis. *In vivo* experiments show that PLA-HA-aPDT can promote *C. auris*-infected wound healing, reduce inflammatory response, and without obvious toxic side-effects. Further results reveal that PLA-HA-aPDT could increase endogenous reactive oxygen species (ROS) levels, leading to mitochondrial dysfunction, release of cytochrome C, activation of metacaspase, and nuclear fragmentation, thereby triggering apoptosis of *C. auris*. Compared with HA, PLA-HA shows stronger controllability and reusability, which can greatly improve the utilization efficiency of HA alone. Taken together, the efficacy, safety and antifungal activity make PLA-HA-aPDT a highly promising antifungal candidate for skin or mucous membrane *C. auris* infection.

## Author summary

It is urgent to develop new antifungal strategies to address the problem of *Candida auris* infection and drug resistance. Previous studies have revealed that antimicrobial

**Funding:** This work was financially sponsored by the National Natural Science Foundation of China (Nos. 81773343 (Y.P.R.), 51803128 (L.T.), 52073186 (L.T.), 81803150 (K.W.Z.). (https://www.nsfc.gov.cn/) The authors also thank the support by the PostDoctor Research Project, West China Hospital, Sichuan University (2020HXBH152 (X.Y.L.)), 1.3.5 project for disciplines of excellence of West China Hospital, Sichuan University (ZYJC18033 (Y.P.R.)), and HX-Academician project of West China Hospital, Sichuan University (HXYS19003 (Y.P.R.)). (http://www.wchscu.cn/index.html, the funder website of West China Hospital, Sichuan University). The funders had no role in the study design, data collection and analysis, decision to publish, or preparation of the manuscript.

**Competing interests:** The authors have declared that no competing interests exist.

photodynamic therapy (aPDT) based on natural products, such as hypocrellin A (HA), is a promising method in clinical applications. However, equivalent studies of aPDT on antifungal activity and its mechanism against *C. auris* remain scarce. Herein, we successfully prepared a recyclable, biodegradable, and light-driven antifungal PLA-HA nanofibrous membrane through the electrospinning technique. *C. auris* infection has been treated by aPDT *in vitro* and *in vivo* for the first time, especially HA-mediated aPDT. *In vitro* and *in vivo* experiments have provided sufficient lines of evidence that PLA-HA is a promising antifungal material for superficial *C. auris* infections due to its antifungal effect and excellent biocompatibility. Notably, there still remains a very high antifungal activity after utilizing PLA-HA four times. In addition, this study clarifies that the anti-*C. auris* mechanism of PLA-HA, namely, PLA-HA-mediated aPDT, is attributed to the formation of intracellular ROS, resulting in mitochondrial dysfunction and a decline in the mitochondrial transmembrane potential, releasing cytochrome C from mitochondria to the cytoplasm, promoting the activation of metacaspase, and inducing nuclear condensation and fragmentation of *C. auris*, thus triggering yeast cell apoptosis. This study lays a foundation for developing new antimicrobial nanofibrous dressings mediated by aPDT and provides an alternative strategy for the treatment of local fungal infectious diseases.

## Introduction

With the widespread application of broad-spectrum antimicrobial agents and immunosuppression, fungal infection has become a major global health threat [1]. The related challenge of the emergence of fungicide resistance is also an increasingly serious problem worldwide [2]. Compared with various antibiotics for bacterial treatment, azoles, polyenes (cell membrane-targeting [2]) and echinocandins (cell wall-targeting [3]) are the only three kinds of antifungal drugs clinically available. The long-term use of antifungal drugs can easily cause the emergence of new and multidrug-resistant strains. *Candida auris*, first isolated from the ear discharge of a female Japanese patient in 2009, is a newly emerged member of the Candida family [4]. As a new opportunistic pathogen, *C. auris* can cause both superficial and life-threatening infections in immunocompromised hosts, especially nosocomial infections. In the past decade, infections caused by *C. auris* have become a global threat due to their rapid emergence worldwide and multidrug resistance properties [5]. Based on the above reasons, *C. auris* is also called "Super fungus", and it has been reported that the proportion of fluconazole-resistant strains even exceeds 90% [6]. Therefore, it is imperative to identify alternative antifungal agents or strategies that are effective against *C. auris* infections.

Antimicrobial photodynamic therapy (aPDT) involves non-toxic photosensitizers (PSs) and suitable light sources to induce the production of reactive oxygen species (ROS), which can kill the microbial pathogens [7,8]. Whether *in vitro* or *in vivo*, numerous studies have demonstrated that ROS can inactivate many microorganisms regardless of the drug-resistance [9–11]. To date, many studies report that aPDT can overcome the resistance in fungi and bacteria, even if microbial pathogens can protect against the damaging effects of ROS and overcome the toxicity of photooxidative stress generated by aPDT to some degree [12–14]. In addition, the type of PS is the decisive factor of aPDT efficiency [15,16].

Hypocrellin A (HA), which is generally regarded as a new type of PS, belongs to the class of perylenequinonoid compounds [17]. This lipid-soluble pigment is one of the main secondary metabolites (SMs) produced by the Chinese medicinal fungus *Shiraia bambusicola* P. Henn. and *Hypocrella bambusae* (Berk. & Broome) Sacc., the fruit bodies of which can unblock

meridians, dissolve phlegm and arrest cough [17,18]. It is well known that HA plays an important role in PDT for anticancer [19], antiviral [20] and antimicrobial [21–23] treatments, due to its production of singlet oxygen and semi-quinone free radicals under light irradiation. However, studies on the antifungal photodynamic activity of HA are limited, only focused on *Candida albicans* [22,24]. As a newly discovered "Super fungus", aPDT or HA treatment on *C. auris* infection has never been reported previously.

Acute and chronic skin wounds are important infective routes for *C. auris* to invade human localized tissue or cause systemic infection [5]. Therefore, anti-infective wound dressings are beneficial to heal wounds and promote tissue repair [25,26]. In antifungal-drug delivery materials, electrospun nano-membranes offer greater air permeability, higher porosity and surface area and higher loading capabilities of PS, making them an excellent choice for withstanding the invasion of exogenous microorganisms [27,28]. Severyukhina *et al.* used chitosan and a second-generation PS to prepare an electrospun nanofiber membrane, and achieved local PDT sterilization with visible light irradiation [29]. In addition, Si *et al.* reported a green bio-based nanofibrous membrane with antimicrobial activity that could be repeatedly sterilized by sunlight excitation [30]. Due to its readily spinnability, drawing-induced crystallization, excellent biocompatibility and biodegradability, the synthetic polymer polylactic acid (PLA) has the largest application in fiber and film manufacturing and biomedical engineering [31,32]. Compared to natural biodegradable polymers, PLA has more controllable mechanical and processible properties [33]. Therefore, PLA has been extensively applied as a substrate in diversified biomedical applications. Electrospinning technology can reconstruct the structure of PLA, transforming it from solid particles into a three-dimensional reticular nanofibrous membrane, which can mimic the architecture and morphology of the extracellular matrix around cells [34]. Moreover, the incorporation of bioactive materials can overcome the insufficiencies and enrich the functions of PLA for particular applications [31].

In this study, we have prepared a novel type of polylactic acid-hypocrellin A (PLA-HA) nanofibrous membrane and tested the PLA-HA based aPDT effects on *C. auris* treatment. First, fungal culture, biofilm construction, morphological observation, and animal models of skin wound infection were used to clarify the role of PLA-HA nanofibrous membrane-aPDT on *C. auris*. Second, the ROS levels and apoptosis hallmarks phosphatidylserine ectropion, mitochondrial transmembrane potential, DNA fragmentation, nuclear condensation, metacaspase and cytochrome C activity in *C. auris* were examined to investigate the cellular mechanism of PLA-HA-aPDT killing (Fig 1). This is the first application of nanofibrous membrane with non-toxic photosensitizer for photodynamic therapy against *C. auris*.

## Results and discussion

### Preparation and characterization of the electrospun nanofibrous membrane

To enhance the stability and controllability of HA, we prepared a PLA-HA nanofibrous membrane (Fig 2). As shown in Fig 2, the color of PLA-HA changed from white (the color of PLA) to red (the color of HA) due to the addition of HA [35]. Micrographs of PLA and PLA-HA nanofibrous membranes were observed by scanning electron microscopy (SEM) (Fig 2), in which the corresponding average fiber diameters were 423 and 699 nm, respectively (S1B and S1C Fig). The surface morphology of fibers presented randomly three-dimensional, uniform and beads-free (Fig 2), which was in accordance with the criteria of an excellent electrospun nanofibrous membrane [36]. After adding HA, the blended solution turned thicker and interactions between molecular chains and solvents increased, which led to an increase in the

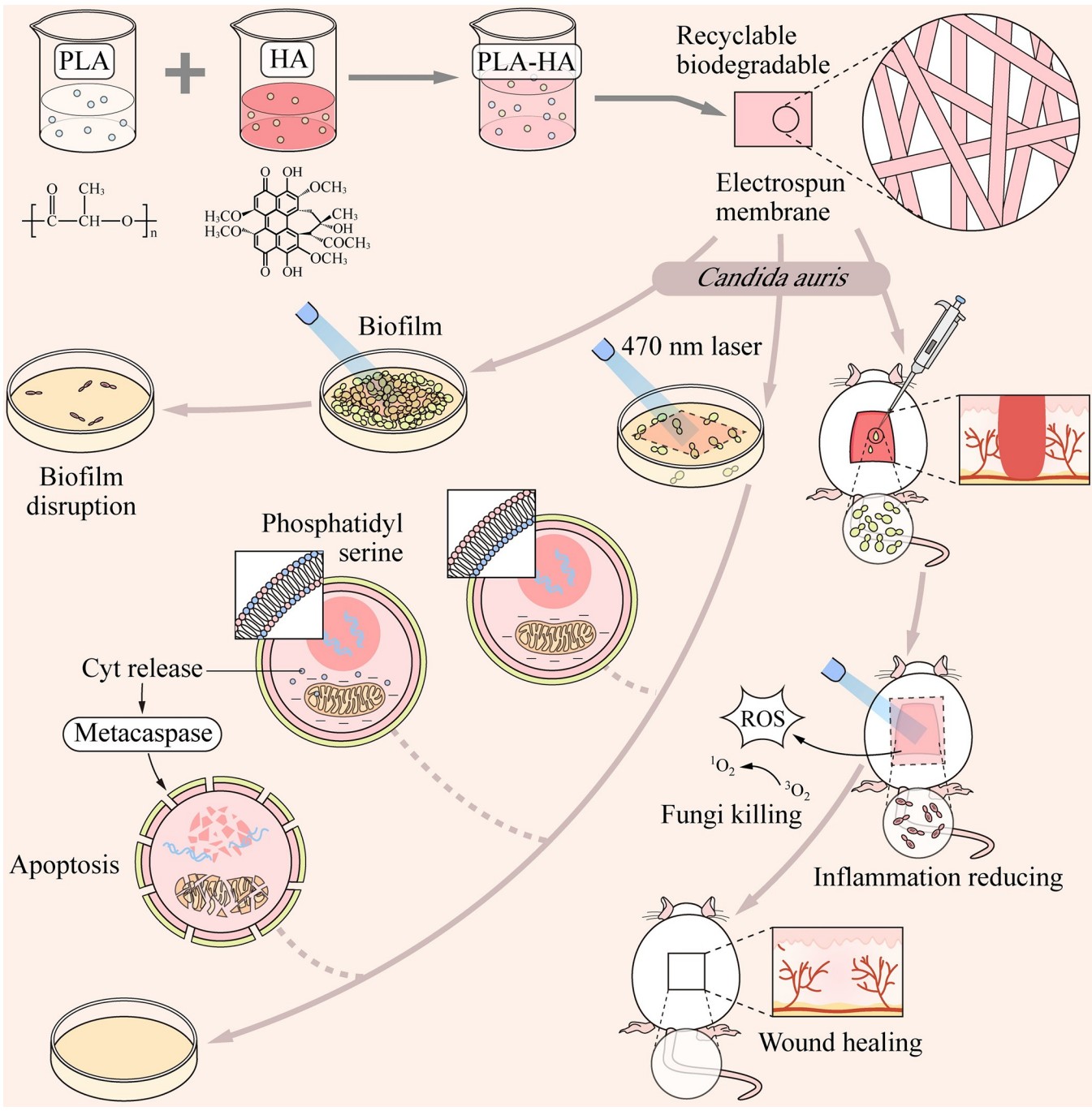

**Fig 1. A schematic illustration of the synthetic route of the PLA-HA membrane and its antifungal applications in *C. auris* planktonic cells and biofilms, as well as the corresponding infected wound healing.**

average diameter of PLA-HA. Other than these changes, we observed no pronounced differences in the structure or morphology of the fibers.

Appropriate mechanical properties should be an important index for ideal wound dressing [37]. S1A Fig exhibits the mechanical properties (breaking force and elongation at break) of the obtained nanofibrous membranes. It was reported that the elongation of human skin varies

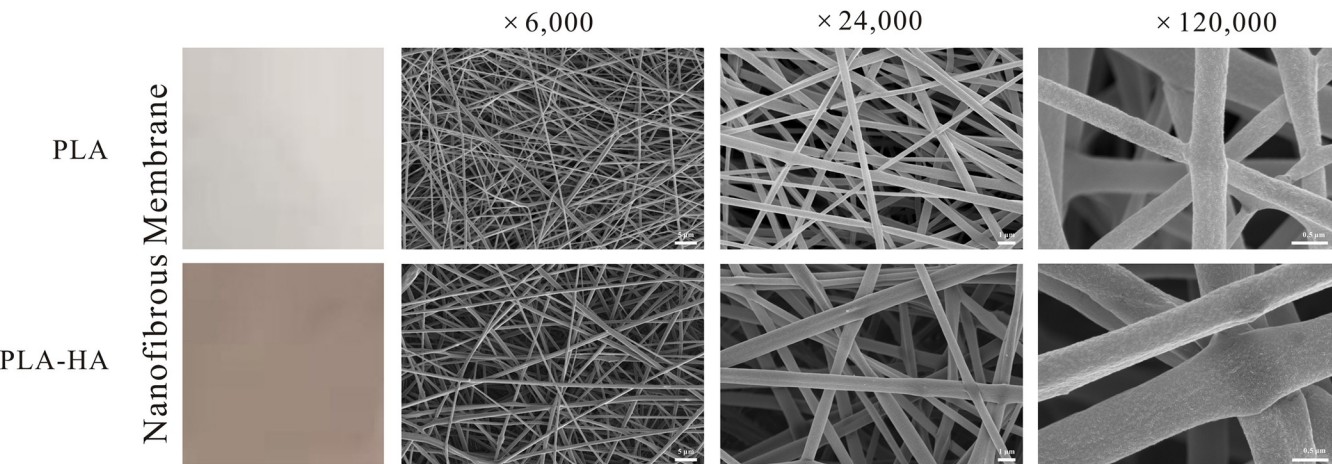

**Fig 2. Visual appearance and surface morphologies of PLA and PLA-HA nanofibrous membranes.** Scale = 5 μm, 2 μm and 0.5 μm from low magnification to high magnification in SEM images.

in the range of 17–207% for elongation at break [38]. In this study, the elongation at break of each membrane was 28.18% for PLA and 33.8% for PLA-HA. Therefore, the mechanical properties of our membranes are in an acceptable elongation range for wound dressings. Taken these together, the increase in the average diameter of PLA-HA led to a decrease in adhesion degree among fibers and an increase in elongation at break, which was consistent with the opinion of the bonding between the components of the membranes exhibiting a significant effect on the mechanical properties [37].

Water vapor transmission rate (WVTR) reflects the capacity of moisture management in wound area of membrane. It is known that high values of WVTR can accelerate the drying process, while a low WVTR would result in the exudates accumulating and microbial infections, hindering the wound healing process [39]. The range of WVTR for human skin is 0.024–0.192 $g \cdot cm^{-2} \cdot 24 \ h^{-1}$, and WVTR of exposed wound is 0.480 $g \cdot cm^{-2} \cdot 24 \ h^{-1}$ [40]. The WVTR of our membranes in this study was in the intermediate. Our results showed that increasing humidity led to a decrease in the WVTR, but the opposite occurred with increasing temperature. In addition, the WVTR of PLA-HA is higher than that of the PLA membrane under the same conditions (S1D and S1E Fig and S1 Table), which indicates that PLA-HA is more beneficial for promoting the volatilization of exudates.

An excellent wound dressing should have the ability to absorb excess exudate from the wound area and maintain a humid environment for wound healing [39]. S1G Fig shows the imbibition rates of PLA (40.5%) and PLA-HA (28.4%) membranes. As revealed in S1G Fig, the addition of HA to the composite decreased the absorption efficiency. Compared with PLA membrane, PLA-HA membrane is more advantageous for reducing the accumulation of exudate on wounds.

In addition, the surface wettability of PLA and PLA-HA membranes was measured by a water contact angle (WCA) system (S1F Fig). The WCA value of the PLA membrane was 97.06˚, while it decreased to 83.36˚ after coating with HA, slightly promoting the hydrophilic property of the membrane specimen.

### *In vitro* antifungal activity

To observe the antifungal activity of PLA-HA, *C. auris* (BJCA001, the first isolated strain in China) was selected [41]. The BJCA001 strain was treated with PLA and PLA-HA with or

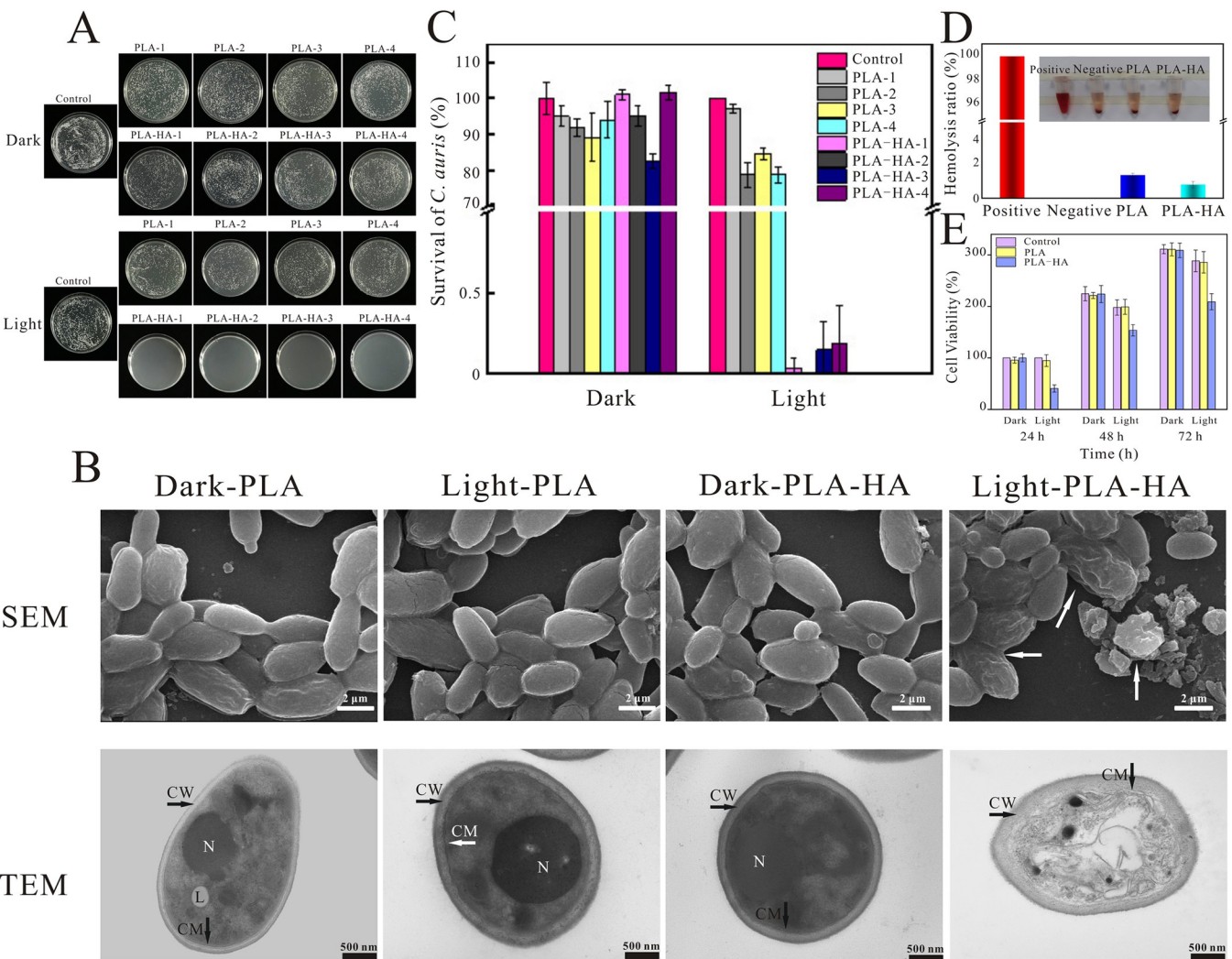

**Fig 3.** ***In vitro*** **antifungal activities and biosafety of PLA-HA and PLA.** (A) Spread plate results of *C. auris* growth under different conditions. The images from left to right represent PLA and PLA-HA were repeatedly used for four times. (B) SEM and TEM images of *C. auris* under different conditions. CM indicates the cytoplasmic membrane, CW represents the cell wall, N indicates the nucleus and L refers to lipid inclusion. (C) Corresponding survival rate results of *C. auris* with PLA and PLA-HA treatment (n = 3). (D) Hemolysis assay. The inset shows the image directly observed after adding PLA and PLA-HA for 2 h. Distilled water was used as a positive control, and PBS was used as a negative control. (E) Cell toxicity evaluation of PLA and PLA-HA on mouse fibroblast L929 cells for 24 h, 48 h and 72 h. Data are presented as the mean ± s.d.

without light irradiation. Agar plate images and quantitative histograms of the colonies are shown in Fig 3A and 3C. Our results show that the PLA and PLA-HA treatments without illumination did not cause any significant toxicity in *C. auris*. However, when illumination was employed for 30 min, the survival rate of the PLA-HA treated strain decreased dramatically, with a survival rate lower than 0.1%. Notably, the antifungal activity of PLA-HA is still very effective after reuse 4 times. Compared with the recently reported antimicrobial materials on HA and its derivatives which fixed on powdery nanocarriers [19,42]. Our material exhibits excellent repeatable antifungal activity.

Moreover, the Live/Dead double staining fluorescent dye Syto9/PI was used to further evaluate the antifungal activity of PLA-HA (S2 Fig). *C. auris* was stained with Syto9 and PI. The Syto9 with green fluorescence could stain all fungal cells, whereas PI only stained cells with

damaged walls. When Syto9 and PI dyes stain a cell at the same time, the fluorescence intensity of Syto9 becomes weaker [43]. As shown in S2 Fig, PLA and PLA-HA treated *C. auris* without illumination, and PLA treated *C. auris* with illumination were stained with bright green fluorescence, indicating that the fungi had an integrated wall and membrane structure. However, the PLA-HA treated strain exhibited red fluorescence, which suggested that the cell wall had been damaged.

SEM and TEM were used to observe how PLA-HA kills *C. auris* (Fig 3B). SEM of *C. auris* shows that the surfaces of the healthy blastospores are smooth with elliptic in shape, and the boundaries between blastospores are clearly visible. However, PLA-HA with illumination could destroy the cell wall and membrane of the fungus, resulting in a shrinkage of cells, and conglutination occurred in the adjacent spores. Additionally, fragments of blastospores are clearly visible in SEM images. From the TEM photographs, the healthy blastospores of *C. auris* in the three control groups showed regular shapes, with a clear outlined cell walls and cytoplasmic membranes. In the cytoplasm, organelles such as nucleus and lipid inclusions were regularly distributed. Meanwhile, the shape of cells of PLA-HA treated *C. auris* is irregular and severely damaged as seen by TEM, the permeability of cell was significantly increased, contents of destroyed fungal flowed out of the cell. Furthermore, some organelles disappeared, some cytoplasms were disintegrated into empty bubbles, the nucleus was fragmented, and some cell walls and membranes were broken and discontinuous.

In addition, the cytocompatibility of the wound dressings plays a pivotal role on wound healing. We used mouse fibroblast (L929) cells to examine the toxicity of PLA and PLA-HA with and without illumination. The results of the CCK-8 assay indicated that neither PLA nor PLA-HA caused obvious cytotoxicity toward L929 cells after 24, 48 and 72 h of incubation without illumination (Fig 3E). However, compared with the control group, the cell viability of L929 cells in the PLA-HA group, declined 59.32%, 22.38% and 27.51% after 24, 48 and 72 h under 470 nm laser irradiation, which means PLA-HA has a slight phototoxicity to L929 cells (Fig 3E).

Hemolysis ratio (HR) is an important factor for the hemo-compatibility of biomaterials. There is no obvious hemolysis of wound dressing when the dressing directly contacted with blood *in vivo* [44]. Therefore, a hemolysis experiment was used to investigate the effect of PLA and PLA-HA on the rupture and lysis of red blood cells (RBCs). As shown in Fig 3D, HR of PLA is 1.51% and the other is 0.89%, indicating that neither PLA nor PLA-HA caused obvious hemolysis and had an excellent blood biocompatibility.

## Capacity of *C. auris* biofilm eradication

Biofilm formation plays an important role in the virulence and pathogenicity of *C. albicans*, which can result in death [45]. Additionally, biofilm formation is often correlated with increased drug resistance among *Candida* species [46]. Fig 4A and 4B show the growth curves of *C. auris* biofilms according to the fluorescence stain assay and XTT reduction method. From Fig 4A, we can see that this strain formed by clumped cells without filamentation. Over time, the gaps between cell clusters of biofilms became narrower within 24 h, indicating biofilm maturation. This result was consistent with the metabolic activity of the biofilm (Fig 4B), which increased rapidly within 24 h but slowly later on.

Therefore, the 24 h biofilm of *C. auris* was selected to assess the biofilm eradication activity of PLA and PLA-HA. As shown in Fig 4C, the inhibition rate of PLA-HA treated biofilms of *C. auris* with illumination was up to 77% compared to the other control groups.

Subsequently, the Live/Dead double staining fluorescent dye Syto9/PI was used to further evaluate the biofilm eradication activity of PLA and PLA-HA; the dyeing principle is

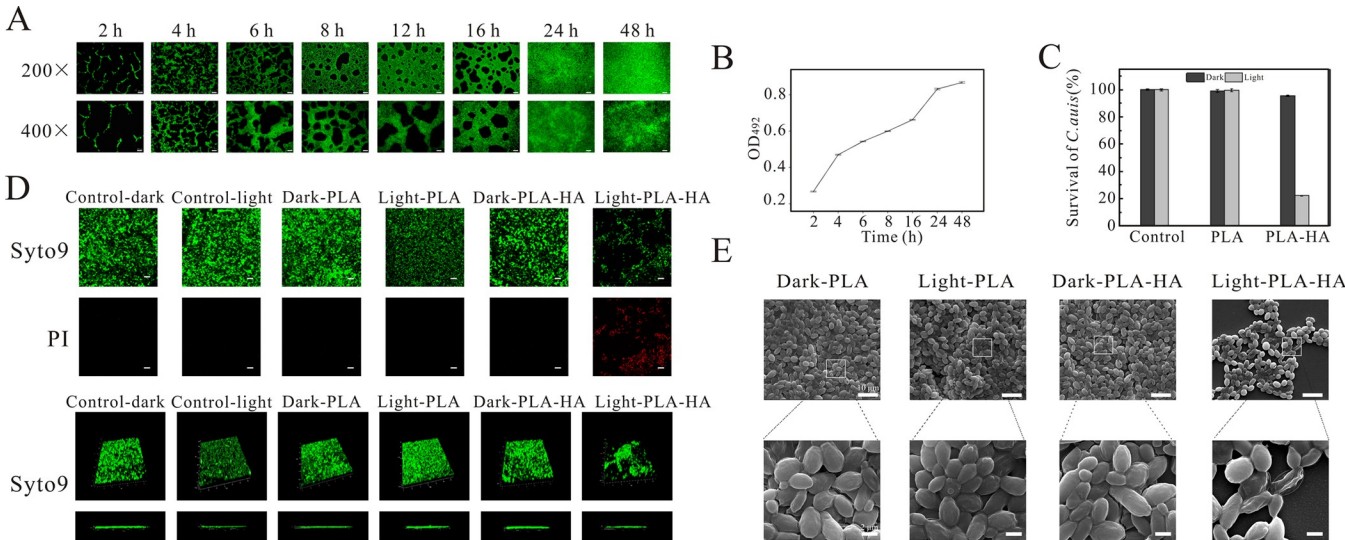

**Fig 4. Biofilm formation and capacity of *C. auris* biofilm eradication.** (A) Observed *C. auris* biofilm formation at different times using Styo9 staining. Scale = 200 μm. (B) Growth curve of *C. auris* biofilm by the XTT reduction method. (C) Capacity of *C. auris* biofilm eradication of PLA and PLA-HA with and without illumination using the XTT reduction method. (D) Live/Dead staining of *C. auris* biofilms observed by CLSM. Scale = 10 μm. E. SEM images of *C. auris* biofilm with PLA and PLA-HA treatment. Data are presented as the mean ± s.d. n = 3.

mentioned above. Compared to the other groups, the PLA-HA treated biofilm of *C. auris* with illumination presented red fluorescence, indicating that the biofilm was badly damaged by CLSM observation (Fig 4D). Fig 4E shows the biofilm eradication activity of PLA and PLA-HA by SEM observation. Compared with the other groups, the biofilm turned sparse and the strains were shrunken after treatment of PLA-HA under a 470 nm laser. Taken together, PLA-HA aPDT could eradicate biofilms of *C. auris*, efficiently.

## *In vivo* antifungal assay

Due to the excellent antifungal ability *in vitro* with no obvious toxicity of PLA-HA to mouse fibroblast cells, we further evaluated the antifungal ability of samples through photodynamic therapy *in vivo* on *C. auris*-infected rats. The infectious and therapeutic processes are shown in Fig 5A. The results showed that the PLA-HA mediated aPDT remarkably promoted wound healing as evidenced by the reduced wound infection size, with the wound healing rate increasing approximately 10% compared to that of the other groups (Fig 5C and 5D). After treatment, the fungi in the infected wound sites of different groups were quantified using the agar plate method. The number of *C. auris* colonies in PLA-HA treated rats was less than 2%, as shown in Fig 5B and 5D, while the fungi grew well in the other groups. This result suggested that the PLA-HA mediated aPDT could decrease *C. auris* burden in the infected rat skin wounds. As shown in S3 Fig, there were no significant differences in white blood cells (WBCs), lymphocytes (LYs), or neutrophils (NEs) between the PLA-HA mediated aPDT group and the other groups. Additionally, the above indexes were in the normal range. This result demonstrated that *C. auris* in our study caused only superficial infection, and not involved systemic infection.

To further determine the anti-infective efficacy of the PLA-HA mediated aPDT, histopathological changes in *C. auris*-infected tissue sites at the end of treatment were examined. H&E staining of skin samples was used to investigate the wound healing process (Fig 5E). The epidermal and dermal structures in the infected tissue in the different groups were damaged at

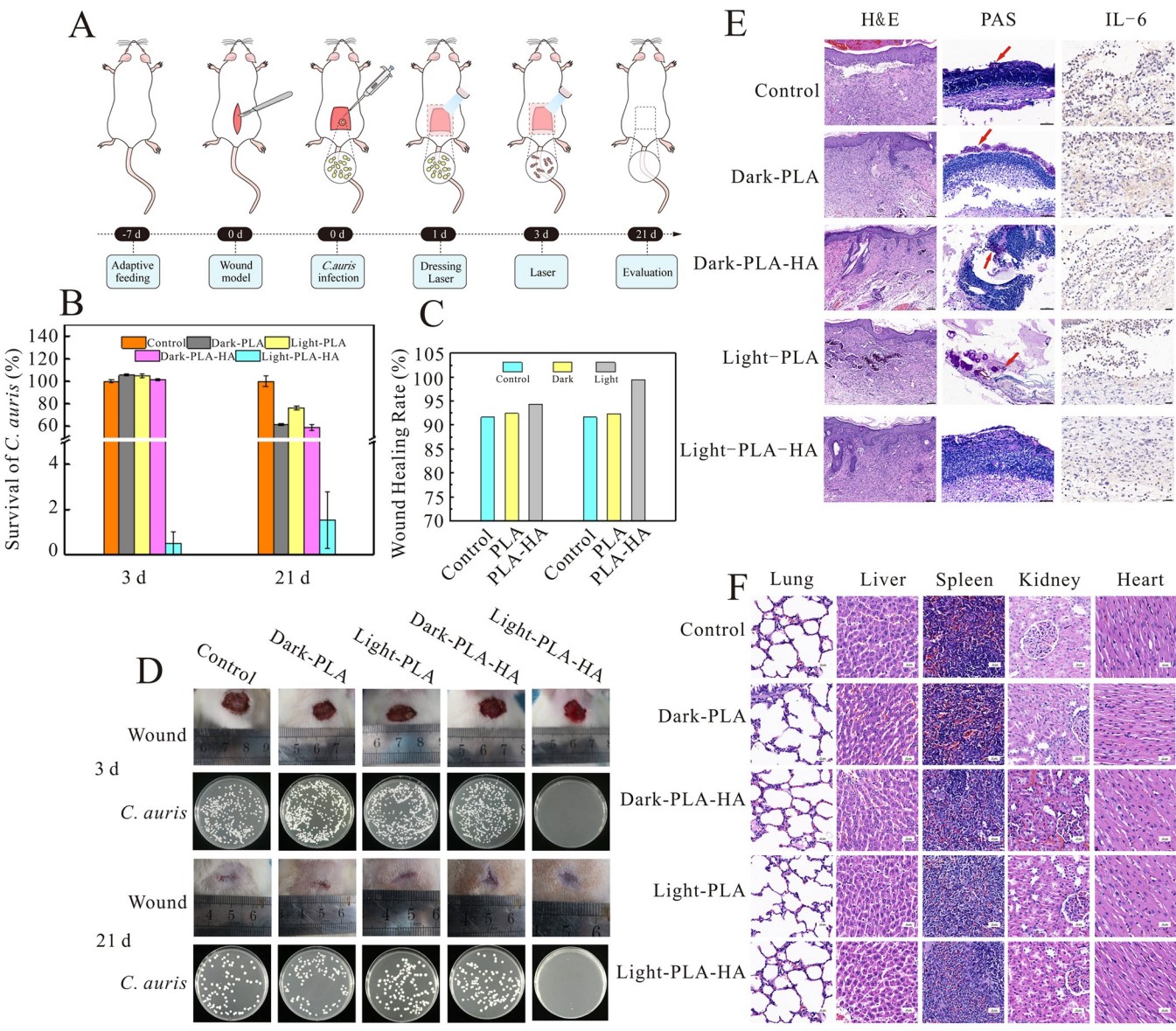

**Fig 5. Antifungal efficacy and compatibility of PLA-HA in treating cutaneous *C. auris* infections.** (A)Schematic diagram of cutaneous wound infection with *C. auris* and treatment with PLA-HA. Rats were evaluated on the 3rd and 21st days. (B) Survival rate of *C. auris* in the infected wound sites treated with PLA-HA and PLA. (C) Corresponding wound healing rate of rats in different groups. (D) Representative photographs of wounds and fungal burden of PLA-HA and PLA treated rats on days 3 and day 21. (E) Infected skin wound tissue evaluated by H&E staining (day 21, scale = 100 μm), PAS staining (day 3, scale = 50 μm) and IHC of IL-6 (day 3, scale = 20 μm) after PLA-HA treatment. (F) H&E staining of lung, liver, spleen, kidney and heart in different groups after treatment (scale = 20 μm). Data are presented as the mean ± s.d.

various levels. In the PLA-HA mediated aPDT group, an intact epidermal layer and some new hair follicles were formed, and a small number of inflammatory cells appeared. In contrast, in the other groups numerous inflammatory cells, local infected tissue necrosis, and bleeding were clearly visible. PAS staining can help to diagnose fungal infection in biopsies and has been considered the most sensitive method for diagnosis [47]. In our study, PAS staining was utilized to observe *C. auris* in tissue sites. The PLA-HA mediated aPDT group presented negative PAS results, but in the other groups the trough-like, circular yeast cells were visible, and the color of the cell wall presented burgundy, which indicated PAS positivity (Fig 5E).

**Table 1. The expression level of IL-6 in *C. auris*-infected tissue sites of different groups.**

| Groups | Average optical density (AOD) |
|---|---|
| Control | 0.019783166 ± 0.0017 |
| Dark-PLA | 0.012174187 ± 0.0035 |
| Dark-PLA-HA | 0.012669057 ± 0.0029 |
| Light-PLA | 0.014977776 ± 0.0026 |
| Light-PLA-HA | 0.001477313 ± 0.0042 |

Data represent the mean ± SD of five replicates.

Furthermore, the expression level of the inflammatory cytokine interleukin-6 (IL-6) in the wound sites was detected by immunohistochemistry (Fig 5E and Table 1). IL-6 positive staining appeared pale brown, and the corresponding average optical density (AOD) could reflect the positive expression level of IL-6, indirectly. In the PLA-HA mediated aPDT group, the expression level of IL-6 was much lower than that in the other groups, indicating that the inflammatory response in rats was significantly controlled by PLA-HA under illumination. All the above experiments proved the strong antifungal activity of PLA-HA mediated aPDT *in vivo*. To further investigate the biocompatibility of PLA and PLA-HA under illumination *in vivo*, the heart, liver, spleen, lung, and kidney of rats were stained with H&E after therapy, and no obvious pathological abnormalities were observed (Fig 5F). This result proved that the PLA-HA mediated aPDT showed no obvious side effects in rats *in vivo*.

## Study of antifungal mechanism

The production of ROS may cause a direct damage to yeast cells since they induce and regulate apoptosis [48]. Therefore, the capacity of generating extracellular and intracellular ROS of PLA-HA directly determines the efficacy of PDT. In this study, the fluorescent probe DCFH-DA, an indicator of ROS, was used to detect intracellular ROS levels (Fig 6A). The production of intracellular $^1O_2$ was further evaluated by singlet oxygen sensor green (SOSG) stained *C. auris*, which can emit a green fluorescence (excitation/emission maxima approximately 504/525 nm) in the presence of singlet oxygen [49]. and the chemical probes DPBF and iodide were utilized to assess the extracellular ROS and $^1O_2$ levels as reported in literature [16,42,50]. As shown in Fig 6B, the PLA-HA aPDT group emitted a strong green fluorescence, while the other groups showed no fluorescence. Fig 6D demonstrated that the fluorescence intensity of DCF in the PLA-HA mediated aPDT group increased at least 80.95% compared with that in the other groups. This result suggests that the PLA-HA mediated aPDT produced ROS accumulation in *C. auris*. As shown in Fig 6C, a strong green fluorescence was detected in the PLA-HA aPDT group, and weak or no fluorescence was observed in the other groups, suggesting that the $^1O_2$ was only produced in *C. auris* in the PLA-HA aPDT group. Fig 6E demonstrated that the absorbance of DPBF decreased rapidly in the PLA-HA mediated aPDT group. The decrease rate of DPBF was proportional to the increase of irradiation time within 30 min, which reached 23% in maximum. This result confirms that the PLA-HA mediated aPDT could efficiently generate extracellular ROS. The extracellular and intracellular results revealed that PLA-HA mediated aPDT possessed a strong ability of generating ROS. On the other hand, iodide has highly specific reaction with $^1O_2$ to form $I^{3-}$ [16], and the oxidation degree was directly proportional to the concentration of $^1O_2$ in an oxygen saturated solution [50]. As shown in Fig 6F, the color of solution changed from white to pale yellow due to intensify of oxidation. When PLA-HA was exposed to a 470 nm laser, two new absorption peaks of 287 and 352 nm appeared and gradually increased linearly in an illumination time-dependent

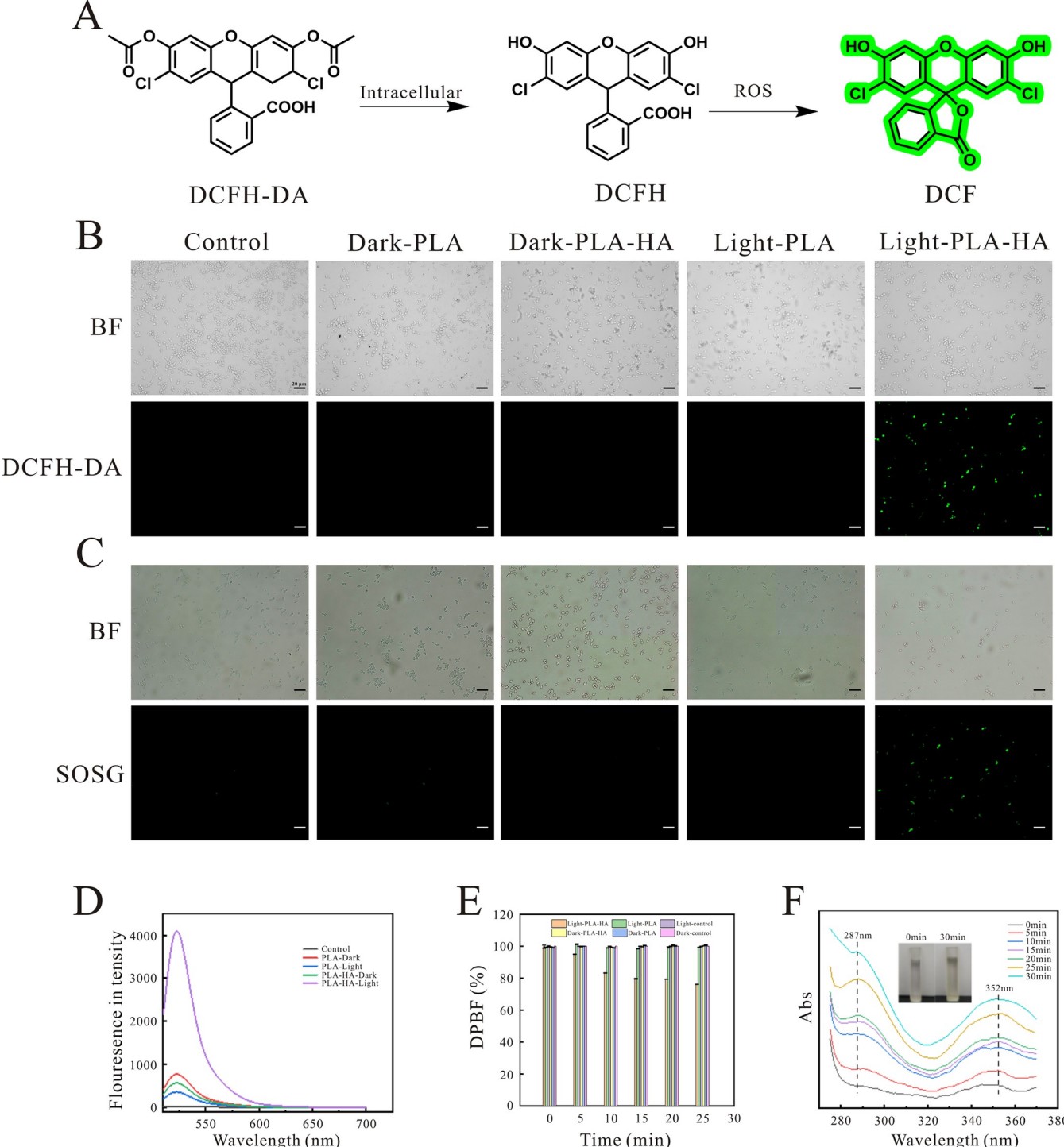

**Fig 6. ROS levels of PLA-HA mediated aPDT.** (A) The detected principle of intracellular ROS. (B) DCFH-DA detected by fluorescence microscopy (scale = 20 μm). (C) Fluorescence microscopy images of SOSG-stained *C. auris* (scale = 20 μm). (D) Changes in intracellular ROS levels after treatment with PLA-HA. (E) Absorbance of DPBF treated with PLA-HA. (F) UV-visible spectroscopic monitoring of the photooxidation of KI to I$^{3-}$ by PLA-HA. Data are presented as the mean ± s.d.

manner because of the increased $I^{3-}$ [50]. This result also demonstrated that PLA-HA mediated aPDT could generate singlet oxygen.

Mitochondria are an important target of ROS. Usually, an abnormal increase in ROS levels would induce the mitochondrial damage and alter the mitochondrial membrane potential, which can be regarded as the first event of apoptosis [51]. In general, high MMP JC-1 accumulates in the mitochondrial matrix to form JC-1 aggregates, emitting red fluorescence. When MMP decreases, JC-1 cannot gather in the mitochondrial matrix, and in the form of JC-1 monomers, which generates a green fluorescence [42]. In our study, the ratio of JC-1 aggregates/monomers was lower in the PLA-HA mediated aPDT treated *C. auris* cells than that in the control groups (Fig 7C), indicating a loss of MMP and the mitochondrial damage.

Next, Annexin V/PI staining and FACS analysis were employed to verify the pathway of cell death induced by the PLA-HA mediated aPDT. Phosphatidylserine (PS) often exists on the inside of the plasma membrane, while in the apoptotic and necrotic cells, PS is exposed on the outside. Thus, this index is regarded as an early marker of apoptosis in fungi [52]. Annexin V can specifically bind to externalized PS, and PI only stains damaged cells [51]. As shown in Fig 7B, the control groups showed a large viable cell population with a few staining for apoptotic cells (about 30% apoptotic cells). Whereas the PLA-HA mediated aPDT treated *C. auris* cells present a large number of apoptotic cell populations (71.94% apoptotic cells), revealing that the PLA-HA mediated aPDT induced *C. auris* apoptosis.

During the early stages of apoptosis, cytochrome C releases from mitochondria to cytoplasm [51]. Thus, the contents of cytochrome C in the mitochondria and cytoplasm were examined in this study. Fig 7D and 7E represent that in the PLA-HA mediated aPDT group, the level of cytosolic cytochrome C distinctly increased, while the mitochondrial cytochrome C content significantly decreased when compared to that in the control cells. This result suggests that the PLA-HA mediated aPDT triggered the release of cytochrome C from mitochondria to cytoplasm.

Furthermore, the release of cytochrome C to cytosol can activate metacaspases, which are caspase-like cysteine proteases in yeast [22]. The activation of metacaspases is a vital part of inducing early stages of apoptosis, and can be detected by FITC-VAD-FMK staining [51]. If the metacaspases are activated, cells present a green fluorescence. Otherwise, the cells appear unstained. Fig 7A shows that the green fluorescence intensity enhanced when *C. auris* were treated with PLA-HA mediated aPDT, suggesting that the PLA-HA mediated aPDT promoted the activation of metacaspase.

Madeo *et al* demonstrated that two pro-apoptotic factors, Aif1 and Nuc1, could result in DNA fragmentation. Additionally, DNA fragmentation is known as a hallmark of apoptosis in the late phase [53]. DNA fragmentation of cells is often detected by the TUNEL assay, which can identify apoptotic DNA cleavage by labeling fluorescent dUTP at the 3'-OH ends of DNA [54]. Fig 7A shows that the fluorescence intensity was remarkably enhanced in the PLA-HA mediated aPDT group, suggesting that this group caused DNA fragmentation. DAPI staining was used to further observe the morphological changes in the nucleus [51]. Compared with the normal nucleus of the control groups, the nucleus of PLA-HA treated yeasts showed a cleaved blue fluorescence or more condensation under a 470 nm laser (Fig 7A), implying that PLA-HA mediated aPDT induced nuclear condensation and fragmentation of *C. auris*.

## Materials and methods

### Ethics statement

This study was approved by the Institutional Animal Care and Ethics Committee of West China Hospital of Sichuan University, with the approval No. 2021621A. Thirty Sprague

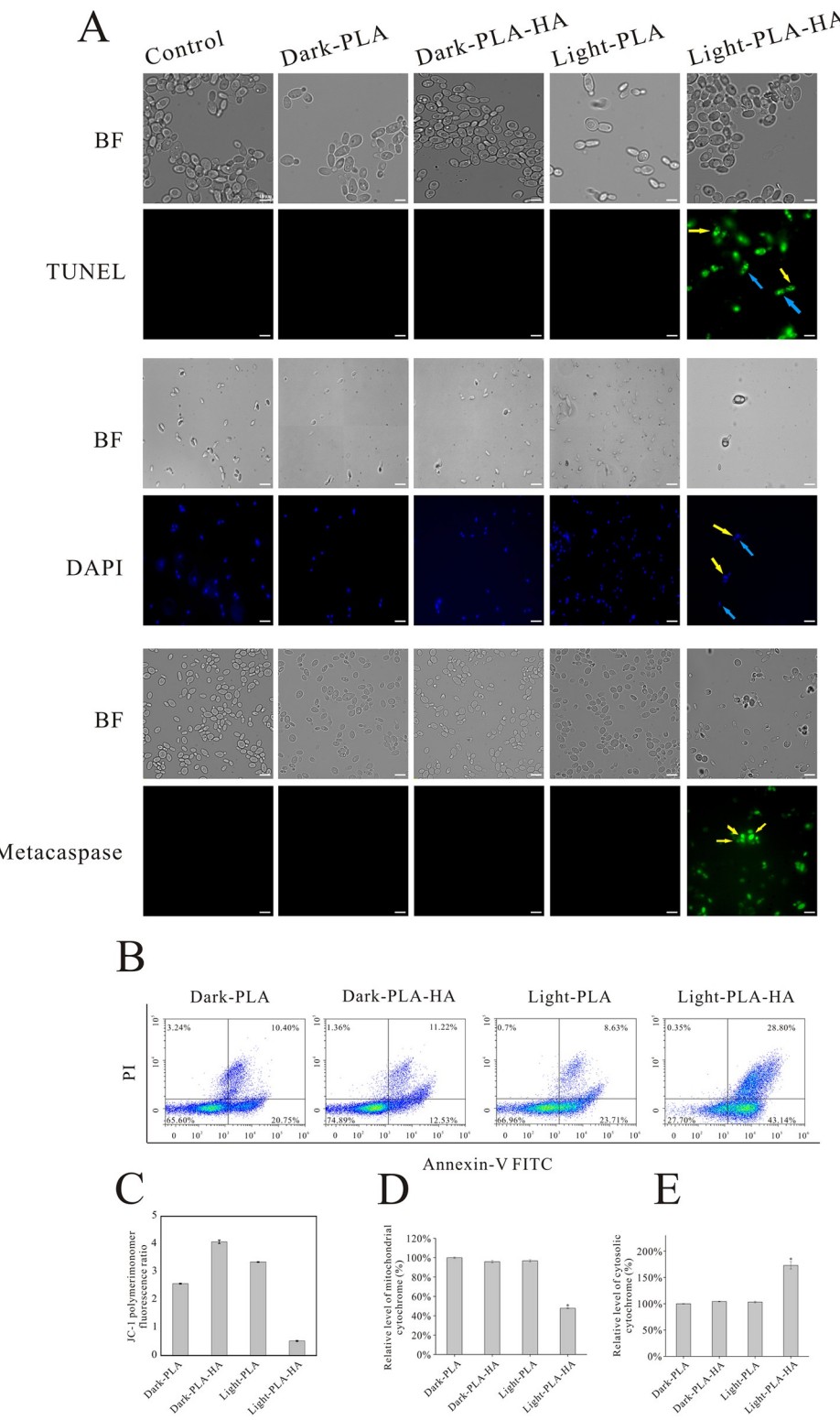

**Fig 7. Antifungal Mechanism of PLA-HA mediated aPDT.** (A) TUNEL, DAPI and Metacaspase detected by fluorescence microscopy. Scale = 10 µm. (B)Apoptosis of *C. auris* cells by staining with Annexin V-FITC and PI. (C) Mitochondrial membrane potential was evaluated using JC-1 staining. (D&E) Cytochrome C release from mitochondria to cytoplasm was assayed after treatment. Data are presented as the mean ± s.d.

Dawley (SD) female rats (180–200 g) were purchased from Chengdu Dashuo Biological Technology Co., Ltd, China. Cages were changed every two weeks to ensure clean bedding, and they had free access to food and water.

## Reagents

All the chemical reagents were directly used without further purification. N, N-dimethyl formamide (DMF), dichloromethane (DCM), PLA, potassium iodide (KI), 1,3- diphenylisobenzofuran (DPBF), and menadione were purchased from Sigma-Aldrich. Absolute alcohol, glutaraldehyde and osmium tetroxide were purchased from Beijing Chemical Reagent Company. An ROS assay kit, JC-1, and one step TUNEL apoptosis assay kit were obtained from the Beyotime Institute of Biotechnology, China. Hypocrellin A (HA, 98%), 2,3-bis(2-methoxy-4-nitro-5-sulfophenyl)-5-[(phenylamino)carbonyl]-2H-tetrazolium hydroxide (XTT) and 3-(N-morpholine) propanesulfonic acid (MOPS) were purchased from Shanghai Yuanye Bio-Technology Co., Ltd.

## Strains and cultures

*C. auris* strain (BJCA001) used in this study was gifted from Professor Guanghua Huang [41]. This strain was routinely grown overnight at 30˚C in yeast peptone dextrose (YPD) medium, which consists of 1% yeast extract (Oxoid, Basingstoke, England), 2% peptone (Solarbio, Beijing, China) and 2% dextrose (Solarbio). For the biofilm of *C. auris* construction, *C. auris* was activated on 9-mm diameter Petri dishes using Sabouraud dextrose agar (SDA), containing 25% dextrose, 1% peptone (Solarbio) and 1.5% agar (Solarbio). The formation of biofilms of *C. auris* was on RPMI 1640 medium, which consisted of 1.04% RPMI 1640 powder (Gibco, USA), 3.45% MOPS and 2% dextrose.

## PLA-HA Mediated aPDT on *C. auris*

In brief, overnight *C. auris* cultures were diluted to ~$10^6$ CFU/mL in YPD medium, and treated with 1 cm × 1 cm nanofibrous membranes (PLA-HA, PLA) for 30 min with or without 470 nm laser (100 mW/cm$^2$) irradiation, respectively. Then, the cultures continued to be incubated for 3 h at 30˚C in the dark. The experiments were divided into four groups: PLA with illumination, PLA-HA with illumination, PLA without illumination and PLA-HA without illumination.

## Antifungal activity assay

The antifungal activities of the nanofibrous membranes were determined by the agar plate dilution method. After aPDT as described above, the treated *C. auris* cells were harvested, and approximately $10^4$ CFU/mL cells were spread onto YPD agar plates. Then, the plates were incubated for 24 h at 30˚C in the dark after spreading. To examine the reutilization of PLA-HA, the nanofibrous membranes were repeatedly used four times. The survival rate was expressed as CFU of the sample group and negative control group. This experiment was independently repeated in triplicate.

## Live/Dead fungal fluorescent imaging

The phototoxicity of nanofibrous membranes to *C. auris* was evaluated by visualized fluorescent imaging. The viability of fungal cells was determined by Syto9 and propidium iodide (PI) dye (Invitrogen Detection Technologies, USA). Syto9 can stain both dead and live fungi with green fluorescence, while PI with red fluorescence only stains dead fungi. Briefly, after aPDT

as described above, fungal cells were washed with PBS and incubated with Syto9 and PI dye (Syto9:PI:PBS = 1.5 μL:1.5 μL:997 μL) in the dark for 30 min at 37˚C. After staining, *C. auris* cells were washed with PBS three times, and 10 μL of cells in PBS were dropped onto a glass slide. Finally, stained cells were imaged using confocal laser scanning microscopy (CLSM, LSM880, Zeiss, Germany).

### Investigating cell morphologies

Briefly, after treatment with nanofibrous membranes, suspensions of *C. auris* were centrifuged at 4000 rpm for 5 min and rinsed with PBS three times. Next, *C. auris* cells were fixed with 2.5% glutaraldehyde at room temperature for 4 h. After being washed with PBS, fungal cells were separately dehydrated by sequential treatment with 20%, 50%, 70%, 80%, 90% and 100% ethanol gradients for 10 min. After critical-point drying they were gilded in a vacuum chamber and observed using scanning electron microscopy (SEM, FEI Inspect F50, equipped with an FEG gun operated at 30 kV at high vacuum).

Transmission electronic microscopy (TEM) was used to observe the inner structure of *C. auris*. The treated *C. auris* cells were fixed with 3% glutaraldehyde overnight at 4˚C in advance, and then fixed in 1% osmium tetroxide for 2 h at room temperature. Next, fungal cells were dehydrated in a series of graded ethanol solutions and propylene oxide and then infiltrated with resin overnight. Subsequently, the semithin sections were stained with methylene blue. After cutting with a diamond knife, ultrathin sections were stained with uranyl acetate and lead citrate. Finally, sections were examined with a JEM-1400-FLASH Transmission Electron Microscope.

### *In vitro* cytotoxicity assay

Cell toxicity of the nanofibrous membranes was evaluated by CCK-8 assay. Nanofibrous membranes (PLA-HA, PLA) were cut into 1 cm × 1 cm size, placed into 24-well plates (n = 5), and washed with PBS three times. Then, L929 cells were seeded into the above 24-well plates at a density of $1.5 \times 10^3$ cells per well, respectively. Besides, the wells without nanofibrous membranes served as the negative control. The experiments were divided into dark groups and light groups (100 mW, 30 min, 470 nm laser). Next, the 24-well plates were placed in the incubator at 37˚C and 5% $CO_2$ for 24, 48 and 72 h. After incubation, the supernatants were removed, and 200 μL CCK-8 solution in serum-free medium was added to the cells. After incubation for another 3 h at 37˚C, 200 μL mixture of each group was transferred to a new 96-well plates. And the absorbance was measured at a wavelength of 450 nm using a microplate reader (SpectraMax M2). The cell viability was calculated using the following equation: Cell viability (%) = $(A_S − A_B) / (A_N − A_B) \times 100\%$. Where $A_S$, $A_N$, and $A_B$ referred to the record absorbance of the sample, negative control, and blank control at a wavelength of 450 nm, respectively.

### Hemolysis assay

Whole blood was collected from rabbits (Dashuo, China) and then added to PBS buffer (pH = 7.4). After centrifugation at 3,500 rpm for 10 min and rinsing with PBS five times, red blood cells (RBCs) were obtained and dispersed in PBS buffer (10% v/v). Then, 1 cm × 1 cm nanofibrous membranes (PLA-HA, PLA) were added to the solution and incubated at 37˚C. After 2 h, the mixtures were centrifuged to examine the hemolysis. RBCs in PBS and distilled water without adding nanofibrous membranes were used as negative and positive controls (n = 5), respectively. The absorbance was measured at a wavelength of 540 nm using a

microplate reader (SpectraMax M2). The hemolysis rate (%) = ($OD_{sample}$ − $OD_{negative\ control}$) /($OD_{positive\ control}$ − $OD_{negative\ control}$) × 100%.

## Biofilm formation

Strains were cultured on SDA plates and incubated at 30˚C for 24 h. A loopful of cultured yeasts was inoculated in 10 mL of YPD broth, and all liquid cultures were kept at 30˚C overnight with shaking (200 rpm). Then, 100 μL seed broth was added to 10 mL fresh YPD broth, and continually incubated to log phase at 30˚C with shaking (200 rpm). Next, cultures were centrifuged at 4000 rpm for 5 min and washed with PBS three times. Then, the washed cells were re-suspended in RPMI 1640 broth medium and the suspension was adjusted to $1 \times 10^6$ cells/mL by blood counting chamber. A total of 200 μL of the suspension was inoculated in flat-bottomed 96-well plates for 2 h at 37˚C. After biofilm preliminary formation, the planktonic cells were discarded, washed with PBS three times. Then, 200 μL RPMI 1640 broth medium was added to 96-well plates and the plates containing preformed biofilms continually incubated for 24 h at 37˚C. Finally, the planktonic cells were discarded, washed with PBS three times. This experiment was independently repeated in triplicate.

## XTT reduction assay

The XTT reduction method is used for quantification of the microbial respiratory activity and evaluation of the toxicity of nanofibrous membranes. Briefly, 1 cm × 1 cm nanofibrous membranes (PLA-HA, PLA) was covered on the biofilms in each well, and were exposed to the 470 nm laser (100 mW/cm$^2$) for 30 min. The groups which kept out of the light served as the negative control. Then, 200 μL of XTT/menadione solution (1 mg/mL XTT,0.4 mM menadione: PBS = 40:2:158) was added to each of the above 96-well plates containing completely dry biofilms (n = 5). The plates were covered with aluminum foil and incubated at 37˚C for 2 h. After incubation, the absorbance was measured at a wavelength of 492 nm using a microplate reader (SpectraMax M2). Blank samples were prepared by filling 1 well with 200 μL of XTT/menadione solution.

## Observation of biofilm

SEM was used to observe the structure of the biofilm, and CLSM was used to observe the Live/ Dead biofilm staining by Syto9 and PI dye. Briefly, 50-mm discs were placed on the bottom of 6-well plates. *C. auris* biofilms were formed by adding 2 mL of yeast suspension in RPMI 1640 broth medium ($1 \times 10^6$ cells/ml) to each well containing discs for 2 h at 37˚C. After discarding the planktonic cells, washing the floating cells with PBS three times, 2 mL fresh RPMI 1640 broth medium was added into the above 6-well plates and continually incubated for 24 h at 37˚C. After incubation, the discs were removed to new 6-well plates and gently washed 3 times with PBS. Subsequently, 1 cm × 1 cm nanofibrous membranes (PLA-HA, PLA) was covered on the disc each well, and were exposed to the 470 nm laser (100 mW/cm$^2$) for 30 min. The groups which were kept out of light were the negative control. Finally, the approach of SEM observation and Live/Dead biofilm were the same as above.

## Generation of extracellular and intracellular ROS and $^1O_2$

1,3-diphenylisobenzofuran probe was used to analyze the levels of extracellular ROS [42]. In brief, nanofibrous membranes (PLA-HA, PLA) were dispersed in 0.5 mg/mL DPBF solution in the dark, then the mixtures were irradiated under a 470 nm laser (100 mW/cm$^2$) for various time periods (0 min, 5 min, 10 min, 15 min, 20 min and 30 min). The generation of ROS was

demonstrated by the characteristic absorption (410 nm) decrease of the DPBF using UV-Vis absorption spectrum.

2,7-dichlorofluorescein diacetate (DCFH-DA) was used to detect the ROS levels in *C. auris*. DCFH-DA itself has no fluorescence, and intracellular esterase can hydrolyze the intracellular DCFH-DA into the membrane-impermeable agent DCFH, which can be rapidly oxidized by ROS to its fluorescent derivative-dichlorofluorescein (DCF). The ROS levels can be reflected by the fluorescence intensities of DCF [42]. Singlet oxygen sensor green (SOSG) was used as a $^1O_2$ probe, which is highly selective for $^1O_2$. This singlet oxygen indicator could emit green fluorescence in the presence of singlet oxygen [49]. In brief, the overnight *C. auris* cultures were diluted to $OD_{600}$ of 0.1 in YPD medium. After incubation to log phase, *C. auris* were treated with nanofibrous membranes (PLA-HA, PLA) for 30 min with or without 470 nm laser (100 mW/cm$^2$) irradiation. Then the *C. auris* were incubated in the dark for the subsequent 3 h. Next, *C. auris* cells were collected, washed thrice with PBS, and stained with 10 μM DCFH-DA or SOSG for 30 min in darkness. After staining, the medium was replaced with PBS. The fluorescence images and green fluorescence intensity of resuspended cells were acquired using a fluorescence microscope (BX43, Olympus, Japan) and fluorescent spectrometer (F-7000, Hitachi, Japan).

## Photooxidation of potassium iodide

KI was chosen as a model substrate to investigate the photooxidation activity of the PLA-HA membrane. In brief, PLA and PLA-HA membranes were cut into 1 cm × 1 cm pieces. Individual membranes were immersed into 20 mL beakers containing 0.1 M KI in the dark, respectively. Then, the mixtures were irradiated under a 470 nm laser (100 mW/cm$^2$) for various time periods (0 min, 5 min, 10 min, 15 min, 20 min and 30 min). The absorption of the solution at 287 nm and 352 nm was recorded by a UV-visible absorption spectrometer.

## Apoptotic detection

The Annexin V-FITC apoptosis detection kit was used to distinguish early/late cellular apoptosis according to the manufacturer's instructions with minor modification. After 470 nm laser induced PDT treatment of nanofibrous membranes as described above, cells were harvested and washed with PBS three times. To obtain protoplasts of *C. auris*, the above cultures ($1 \times 10^6$ CFU/mL) were treated with 2% snailase for 2 h at 30˚C. After removing cell walls, the cultures were centrifuged gently, washed with sorbitol (1.2 M) three times, and then resuspended in binding buffer (containing 1.2 M sorbitol). To assess the cellular integrity and the externalization of phosphatidylserine (PS), protoplasts of *C. auris* were stained with Annexin V-FITC and PI for 15 min in the dark at room temperature. Next, the induction of apoptosis was determined by FACS Caliber flow cytometer (FACSCalibur; Becton Dickinson, San Jose, CA). A total of 10,000 events were counted at the flow rate. Data analysis was performed using Cell Quest software (Becton Dickinson Immunocytometry Systems). All experiments were performed in triplicate.

## Mitochondrial membrane potential assay

As the hallmark for early cellular apoptosis, mitochondrial membrane potential (MMP) was measured using 5,5',6,6'-tetrachloro-1,1',3,3'-tetraethyl-benzimidazolyl-carbocyanine iodide (JC-1) by detecting a switch from red to green fluorescence. After aPDT as described above, the treated *C. auris* cells were harvested, washed, and stained with 2.5 μg/ml JC-1 for 30 min in the dark. After washing with PBS three times, the cells were analyzed by a fluorescence

spectrophotometer (F-7000, Hitachi, Japan). The ratio of the fluorescence intensities of JC-1 aggregates to JC-1 monomers was calculated.

## Detection of nuclear condensation

Nuclear condensation was examined using 4',6-diamidino-2-phenylindole (DAPI) staining. First, the cells in the four groups were treated as described above. Second, the treated cells were collected and washed with PBS three times. Then, the cells were fixed in 70% ethanol for 1 h on ice. After that, the cells were washed with PBS three times and stained with 2 μg/ml DAPI for 30 min in the dark. Finally, the cells were washed and observed by fluorescence microscopy.

## Determination of DNA fragmentation

The terminal deoxynucleotidyl transferase (TdT) dUTP nick end labeling (TUNEL) staining is widely used for detecting cell apoptosis. After aPDT as described above, the treated yeast cells were collected and washed with PBS three times. Then, the cells were fixed in 3.6% paraformaldehyde for 30 min, and permeabilized in PBS which containing 0.3% Triton X-100 on ice for 2 min. After that, the cells were washed with PBS three times and stained with a one-step TUNEL assay kit for 1 h at 37˚C. Finally, the apoptotic cells were assessed by fluorescence microscopy.

## Metacaspase activation assay

CaspACE FITCVAD-FMK in situ marker (Promega) was used to detect the metacaspase activities according to the manufacturer's protocols. After 470 nm laser triggered aPDT of nanofibrous membranes as described above, the treated *C. auris* cells were collected and washed with PBS three times. Then, the cells were stained with 5 μM CaspACE FITC-VAD-FMK at 30˚C for 30 min in darkness. Finally, the cells were washed again and analyzed by fluorescence microscopy.

## Quantification of cytochrome C in cytoplasm and mitochondrion

Cytochrome C levels were examined as described previously with minor modification [51]. After aPDT as described above, the treated *C. auris* were harvested. Then, the protoplasts were prepared with 2% snailase, according to the above methods. After that, the protoplasts were homogenized in buffer A (50 mM Tris, 2 mM EDTA, 1 mM phenylmethylsulfonyl fluoride, pH = 7.5), and the supernatant was collected. Next, the obtained supernatant was centrifuged at 50,000 rpm for 45 min. The supernatant was used to measure the cytochrome C releasing to cytoplasm. On the other hand, the pellet was suspended in buffer B (50 mM Tris, 2 mM EDTA, pH = 5.0) to obtain pure mitochondria. Finally, after treatment with 500 mg/mL ascorbic acid for 5 min, the cytochrome C content in the cytoplasmic or mitochondrial samples was quantified at a wavelength of 550 nm using a microplate reader (SpectraMax M2).

## Rat model of skin wound infection

All animal experiments were approved by the Institutional Animal Care and Ethics Committee of West China Hospital of Sichuan University (Approval No. 2021621A). Thirty Sprague Dawley (SD) female rats (180–200 g) were purchased from Chengdu Dashuo Biological Technology Co., Ltd, China. Before surgery, the SD rats were adjustable fed for one week to adapt the environment. The rats were anesthetized by intraperitoneal injection of chloral hydrate (10%, 3 mL/kg). The dorsal hair of the rats was removed using a razor and depilatory cream. Then, a

full-thickness excisional skin wound (diameter: 20 mm) was created on the dorsum of each rat under sterile conditions. A suspension of *C. auris* (60 μL, $10^6$ CFUs) was inoculated into each wound. After that, all rats were kept in separate cages and raised at 25˚C for 2 days. Then, the rats were placed into four groups: *C. auris* infection + PLA-HA membrane + 470 nm illumination group, *C. auris* infection + PLA membrane + 470 nm illumination group, *C. auris* infection + PLA-HA membrane group, and *C. auris* infection + PLA membrane group (n = 6). The observation of *C. auris* infection on the rat wounds was conducted on day 3 and day 21. To measure the fungal burden, the rat wounds were surgically excised, weighed, and homogenized in PBS. Then, the suspensions containing tissues and *C. auris* were subjected to serial 10-fold dilutions and then spread onto YPD agar plates containing streptomycin and ampicillin. After 48 h incubation at 30˚C, the number of colonies was counted. To evaluate the wound contraction rate, a digital camera (EOS 1300D, Canon, Japan) was applied to capture the images of wound beds during the whole healing process. The wound area of each wound was analyzed using *Image J* software.

## Histopathology assessment

Briefly, wound tissue and organs from rats in different groups were fixed in formalin solution and embedded in paraffin. Then, tissues were sectioned into 3 μm thick sections. After that, tissue sections were dewaxed in xylene and gradient dehydrated using a graded series of ethanol. Finally, sections were stained with hematoxylin and eosin (H&E) and scanned using Pannoramic 250 (3DHISTECH, Hungary). For the periodic acid-Schiff (PAS) staining, another set of paraffin blocks were rehydrated using the same method, but they were stained with 1% periodic acid dye (Scy Tech, USA). For the immunohistochemical assay, wound tissue sections were labeled using IL-6 antibodies. Briefly, after dewaxing and rehydration, sections were washed with PBS and incubated with 1% hydrogen peroxide for 15 min to deplete endogenous peroxide. Next, sections were placed in sodium citrate buffer, heated in a microwave at 140˚C for 3 min, and cooled for 15 min at room temperature. Next, sections were blocked with goat serum at 37˚C for 20 min and subsequently incubated with primary antibodies against IL-6 at 4˚C overnight. Next, sections were washed with PBS, and horseradish peroxidase (HRP) -conjugated secondary antibody (dilution ratio, 1:100) was added for 1 h at room temperature, the immune reaction was visualized by color development with DAB. Finally, sections were counterstained with hematoxylin and scanned using Pannoramic 250. Under the microscope magnifying 400 times, 5 sections of each sample in 3 random sights were selected, and the Image-Pro Plus 6.0 (Media Cybernetics, Inc., Rockville, MD, USA) was used to conduct the semi-quantitative analysis of integrated optical density (IOD) and pixel area (AREA), the average optical density (AOD) was the ratio of IOD and AREA.

## Statistical analysis

The results were analyzed by SPSS software. Every experiment was independently performed at least three times, and the data were expressed as the mean ± SD.

## Supporting information

**S1 Fig. Characterization of nanofibrous membranes.** (A) The mechanical properties of PLA and PLA-HA. The relationship between the breaking strength and elongation at break of PLA and PLA-HA was detected. (B&C) The diameter distributions of PLA-HA and PLA. (D&E) WVTR of PLA-HA and PLA with different detected humidities and temperatures. (F) Water contact angle measurements of PLA and PLA-HA. (G) The imbibition rate of PLA and

PLA-HA. Data are presented as the mean ± s.d.
(TIF)

**S2 Fig. Live/Dead of *C. auris* using Syto9 and PI staining, observed by CLSM.**
(TIF)

**S3 Fig. Routine blood examination of *C. auris* after treatment.**
(TIF)

**S1 Table. WVTR of PLA-HA and PLA with different detected humidities and temperatures.**
(DOCX)

**S1 Data. Excel spreadsheet containing, in separate sheets, the underlying numerical data and statistical analysis for Figs 3C, 3E, 4B, 4C, 5B, 6E, 7C, 7D, 7E, S1A, S1B, S1D, S1E and S1F.**
(XLSX)

**S1 Text. The methods of preparation and characterization of electrospun nanofibrous membranes.**
(DOC)

## Acknowledgments

We thank Prof. Guanghua Huang, Fudan University, for the kind supply of *C. auris* strain BJCA001. The authors would like to thank Senior Experimentalist Chaoliang Zhang, State Key Laboratory of Oral Diseases, West China Hospital of Stomatology, Sichuan University, for conducting SEM observations.

## Author Contributions

**Conceptualization:** Xinyao Liu, Wei Chen, Lin Tan, Yuping Ran.

**Data curation:** Xinyao Liu, Chuan Guo.

**Formal analysis:** Xinyao Liu, Chuan Guo, Kaiwen Zhuang, Lin Tan.

**Funding acquisition:** Xinyao Liu, Kaiwen Zhuang, Lin Tan, Yuping Ran.

**Investigation:** Xinyao Liu, Chuan Guo, Muqiu Zhang, Yalin Dai.

**Methodology:** Xinyao Liu.

**Project administration:** Xinyao Liu, Chuan Guo.

**Supervision:** Lin Tan, Yuping Ran.

**Validation:** Xinyao Liu, Muqiu Zhang, Yalin Dai.

**Visualization:** Xinyao Liu.

**Writing – original draft:** Xinyao Liu.

**Writing – review & editing:** Wei Chen, Lin Tan, Yuping Ran.

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
