## [Decision Letter · Decision Letter 0]

21 Mar 2022

Dear Professor Ran,

Thank you very much for submitting your manuscript "A recyclable and light-triggered nanofibrous membrane against the emerging fungal pathogen Candida auris" for consideration at PLOS Pathogens. As with all papers reviewed by the journal, your manuscript was reviewed by members of the editorial board and by several independent reviewers. In light of the reviews (below this email), we would like to invite the resubmission of a significantly-revised version that takes into account the reviewers' comments.  You will note that both reviewers have recommended additional experiments (or controls) be performed.  A third potential reviewer also looked over the paper and was concerned by the quality of the language.  A tracked changed editor version has been supplied, but it would be wise to check the text carefully.  One last point (raised by reviewer 1) is that the figures in many cases are small; the journal will have an online version but also PDF that some people will print at the set size.  It may be redesign, or splitting some figures into multiples, would make the data obtained more clearly visualised.

We cannot make any decision about publication until we have seen the revised manuscript and your response to the reviewers' comments. Your revised manuscript is also likely to be sent to reviewers for further evaluation.

Sincerely,

Alexander Idnurm

Guest Editor

PLOS Pathogens

Xiaorong Lin

Section Editor

PLOS Pathogens

Kasturi Haldar

Editor-in-Chief

PLOS Pathogens

orcid.org/0000-0001-5065-158X

Michael Malim

Editor-in-Chief

PLOS Pathogens

orcid.org/0000-0002-7699-2064

Reviewer's Responses to Questions

**Part I - Summary**

Reviewer #1: This manuscript developed a light-triggered PLA-HA nanofibrous membrane against C. auris. The authors found PLA-HA-aPDT can promote the C. auris -infected wound healing, reduce inflammatory response without obvious toxic side-effects, increase endogenous ROS levels, leading to mitochondrial dysfunction, release of cytochrome C, activation of metacaspase, and nuclear fragmentation, thereby triggering an apoptosis of C. auris. The manuscript is well structured, the characterization is comprehensive and helpful for the readers who are working on antifungal research. I would recommend the publication of this manuscript after addressing the following comments.

Reviewer #2: This manuscript is interesting to use a recyclable and biodegradable polylactic acid-hypocrellin A (PLA-HA) nanofibrous membrane to treat C. auris. Although its antifungal activity and the involved mechanism against C. auris is similar to the previous studies by Yang et al. (2019). This is manuscript is still meaningful to publish.

**Part II – Major Issues: Key Experiments Required for Acceptance**

Reviewer #1: (No Response)

Reviewer #2: 1. The authors said that PLA-HA mediated aPDT could generate singlet oxygen. Then whether PLA-HA-induced C. auris apoptosis is due to singlet oxygen, which needs to be investigated.

**Part III – Minor Issues: Editorial and Data Presentation Modifications**

Reviewer #1: 1) Introduction: The authors reviewed C. auris, PS, HA, nanofibrous membrane. However, there is a lack of review on the effect of PLA against fungi, and its advantages and disadvantages as nanomembranes. Related literatures (e.g., Composites Part B: Engineering, 2020, 199, 108238; PLoS Pathogens, 2021,17, e1009470.) should be cited in the introduction considering the advances.

2) Figure 3e, the examination of the toxicity of PLA and PLA-HA using mouse fibroblasts was performed in the absence of light. I suggest the authors clarify the effect of light on toxicity.

3) The insets in the SEM images of Figure 4e are not clear.

4) ‘Fig 4e shows ... Compared with other groups, the biofilm turned sparse and the strains were shrunken after treatment of PLA-HA.’ The authors are suggested to provide SEM images with lower magnification in the supplementary file to support this conclusion.

Reviewer #2: 1.Please describe the advantage of PLA-HA compared with HA.

PLOS authors have the option to publish the peer review history of their article (what does this mean?). If published, this will include your full peer review and any attached files.

Reviewer #1: No

Reviewer #2: **Yes: **Chang Jia

Editor comments:

See comments in the text above and also the attached word document.
---

## [Editor Report · Decision Letter 1]

20 Apr 2022

Dear Professor Ran,

We are pleased to inform you that your manuscript 'A recyclable and light-triggered nanofibrous membrane against the emerging fungal pathogen Candida auris' has been provisionally accepted for publication in PLOS Pathogens.

IMPORTANT: The editorial review process is now complete. PLOS will only permit corrections to spelling, formatting or significant scientific errors from this point onwards. Nonetheless, please make sure that one last careful read over the text is performed to avoid anything carrying over to the final manuscript.  Requests for major changes, or any which affect the scientific understanding of your work, will cause delays to the publication date of your manuscript.

Best regards,

Alexander Idnurm

Guest Editor

PLOS Pathogens

Xiaorong Lin

Section Editor

PLOS Pathogens

Kasturi Haldar

Editor-in-Chief

PLOS Pathogens

orcid.org/0000-0001-5065-158X

Michael Malim

Editor-in-Chief

PLOS Pathogens

orcid.org/0000-0002-7699-2064
---

## [Editor Report · Acceptance letter]

20 May 2022

Dear Professor Ran,

We are delighted to inform you that your manuscript, "A recyclable and light-triggered nanofibrous membrane against the emerging fungal pathogen *Candida auris* ," has been formally accepted for publication in PLOS Pathogens.

Best regards,

Kasturi Haldar

Editor-in-Chief

PLOS Pathogens

orcid.org/0000-0001-5065-158X

Michael Malim

Editor-in-Chief

PLOS Pathogens

orcid.org/0000-0002-7699-2064